# Enhancing Parameter Efficiency and Generalization in Large Models: A Regularized and Masked Low-Rank Adaptation Approach

**Yuzhu Mao**      *myz20@mails.tsinghua.edu.cn*
**Zihao Zhao**      *zhao-zh21@mails.tsinghua.edu.cn*
**Siqi Ping**      *psq22@mails.tsinghua.edu.cn*
*Tsinghua-Berkeley Shenzhen Institute*
*Tsinghua Shenzhen International Graduate School, Tsinghua University*
**Yang Liu**      *liuy03@air.tsinghua.edu.cn*
*Institute for AI Industry Research (AIR), Tsinghua University*
*Shanghai Artificial Intelligence Laboratory*
**Wenbo Ding**[*]      *ding.wenbo@sz.tsinghua.edu.cn*
*Tsinghua-Berkeley Shenzhen Institute*
*Tsinghua Shenzhen International Graduate School, Tsinghua University*
*Shanghai Artificial Intelligence Laboratory*

*Reviewed on OpenReview:* *https://openreview.net/forum?id=zgOhPlABfY*

## Abstract

Large pre-trained models, such as large language models (LLMs), present significant resource challenges for fine-tuning due to their extensive parameter sizes, especially for applications in mobile systems. To address this, Low-Rank Adaptation (LoRA) has been developed to reduce resource consumption while maintaining satisfactory fine-tuning results. Despite its effectiveness, the original LoRA method faces the challenge of suboptimal performance. This paper investigates the intrinsic dimension of the matrix updates approximated by the LoRA method and reveals the performance benefits of increasing this intrinsic dimension. By employing regularization and a gradient masking method that encourages higher intrinsic dimension, the proposed method, termed **R**egularized and **M**asked LoRA (RM-LoRA), achieves superior generalization performance with the same or lower trainable parameter budget compared to the original LoRA and its latest variants across various open-source vision and language datasets.

## 1 Introduction

Large pre-trained models, such as large-scale language models (LLMs), have showcased remarkable performance across a variety of tasks in computer vision and natural language processing (Zhang et al., 2022; Brown et al., 2020; Touvron et al., 2023; Ouyang et al., 2022). Nonetheless, fine-tuning these models for specific downstream tasks often presents substantial resource challenges due to their extensive parameter sizes. In this context, parameter-efficient fine-tuning (PEFT) methods have been extensively explored to alleviate resource consumption while preserving or enhancing fine-tuned model performance (Zaken et al., 2022; Hu et al., 2021; Li & Liang, 2021; Lester et al., 2021; Vu et al., 2022; Guo et al., 2021; Zhang et al., 2023b; Liu et al., 2022a; Houlsby et al., 2019; Liu et al., 2022b; Sung et al., 2021; 2022; Mao et al., 2022; Lee et al., 2020). Among these methods, Low-Rank Adaptation (LoRA), which involves freezing the pre-trained weights and approximating updates in weight matrices using the multiplication of two low-rank matrices, has emerged

---

[*]Corresponding Author

as a promising approach to balance computational efficiency and task performance during the fine-tuning process (Hu et al., 2021).

Despite its effectiveness, LoRA fine-tuning encounters challenges in determining the optimal size of LoRA matrices for a given model and task. On one hand, excessively small matrices, with limited number of trainable parameters, inevitably harm training convergence and generalization performance. On the other hand, large matrices introduce redundant trainable parameters, which could be reduced to enhance parameter efficiency. Moreover, some studies have indicated that large LoRA matrices may exacerbate overfitting, as redundant parameters primarily contribute to training accuracy rather than test accuracy (Qiang et al., 2024; Karimi Mahabadi et al., 2021).

Several approaches have been proposed to determine or adaptively adjust the size of LoRA matrices, often referred to as the LoRA rank $R$, for improved efficiency and generalization (Valipour et al., 2023; Benedek & Wolf, 2024; Kopiczko et al., 2023; Ding et al., 2023; Zhang et al., 2023a). However, none of these methods investigate the intrinsic dimension $r$ of the approximated matrix update $\Delta \mathbf{W} = \mathbf{BA}$ given by the product of low-rank LoRA matrices $\mathbf{A}$ and $\mathbf{B}$. This intrinsic dimension $r$, instead of the previously studied LoRA rank $R$, has been proven to play a crucial role in LoRA fine-tuning. Specifically, it has been theoretically demonstrated that for fully connected neural networks, the LoRA approximation error given by an approximated update $\Delta \mathbf{W} = \mathbf{BA}$ with intrinsic dimension $r$ is related to the $r$-th singular value of the discrepancy $\mathbf{E} = \mathbf{W}_{\text{target}} - \mathbf{W}_{\text{frozen}}$ between the target weight matrix and the frozen pre-trained weight matrix (Zeng & Lee, 2023). In this context, the previously studied LoRA rank $R$ determined by the size of matrices $\mathbf{A}$ and $\mathbf{B}$ acts as just an upper bound for $\Delta \mathbf{W}$'s intrinsic dimension $r$.

In other words, encouraging the intrinsic dimension $r$ of $\Delta \mathbf{W}$ to approximate the given LoRA rank $R$ benefits the generalization of LoRA fine-tuning under a given trainable parameter budget. Inspired by this theoretical conclusion, this paper first adopts a regularization technique to encourage LoRA matrices to span a higher intrinsic rank in their parameter space. Additionally, to maintain a reasonable budget of trainable parameters, a gradient masking method is introduced to randomly mask a subset of parameters in each epoch instead of updating all parameters in LoRA matrices. Experiments on multiple datasets have proven this method also helps promote the growth of the intrinsic rank $r$ and thus yields lower approximation error and better generalization performance.

The contributions of this paper can be summarized as follows:

1. This paper extends previous theoretical bounds for LoRA approximation error from simulated datasets to real-world datasets, providing further insights into the trade-off between LoRA rank $R$ and generalization performance.

2. Based on the analysis of LoRA rank and generalization performance, this paper designs a strategy for fine-tuning LoRA matrices that encourages the growth of intrinsic rank $r = \text{rank}(\Delta \mathbf{W})$ within the LoRA parameter space defined by $R$. This strategy effectively alleviates the problem of overfitting the training data by encouraging the LoRA matrices to explore the parameter space.

3. The experimental results across multiple open-source datasets demonstrate that this **R**egularized and **M**asked version of LoRA (**RM-LoRA**) method manages to strike a better efficiency-generalization tradeoff compared to the original LoRA method and its state-of-the-art variations, with better generalization performance achieved with the same or lower trainable parameters budget.

## 2 Related Works

In attempts to address the computational challenges posed by updating the enormous amount of weights in large pre-trained models, LoRA achieves outstanding model generalization with a significantly reduced budget of trainable parameters during fine-tuning (Hu et al., 2021). However, LoRA still faces the challenges of sub-optimal performance. Previous research addressing LoRA's main challenges is briefly discussed as follows:

**Underfitting**. While LoRA has demonstrated remarkable parameter efficiency and generalization performance, it may lead to insufficient fine-tuning of large-scale models with high embedding dimensions (Hayou

et al., 2024). In some cases, there is a contradictory phenomenon where a higher LoRA rank doesn't necessarily yield better training results than a lower LoRA rank. Much research has been devoted to further enhancing both the performance and efficiency achieved by LoRA, including the adaptive choice of LoRA rank (Zhang et al., 2023a; Ding et al., 2023; Valipour et al., 2023), adjustment of learning rate (Hayou et al., 2024), random projection (Kopiczko et al., 2023), derivative-free optimization (Jin et al., 2024), and pre-trained weights optimization (Zi et al., 2023). Nevertheless, none of these methods consider the role of LoRA updates' intrinsic dimension in mitigating the performance gap under a given LoRA rank setting.

**Overfitting.** Fine-tuning large pre-trained models with a large number of parameters can easily lead to overfitting (Karimi Mahabadi et al., 2021). In the AdaLoRA method, the LoRA matrices of less important pre-trained weight matrices are assigned a lower rank to prevent overfitting (Zhang et al., 2023a). However, according to the experiments by Qiang et al. (2024), LoRA and AdaLoRA still clearly overfit the training data as fine-tuning progresses, with decreases in training losses but increases in test losses. To alleviate the overfitting problem, Qiang et al. (2024) developed the BiLoRA method, which iteratively trains different subsets of trainable parameters using different subsets of training data. Furthermore, Hayou et al. (2024) pointed out that the overfitting of LoRA and its variants is due to some directions of LoRA matrices not being sufficiently updated, and thus the change in model weights approximated by LoRA is restricted by the vector (sub)space generated by the LoRA matrices' columns at initialization.

The previous analysis of LoRA's existing limitations and solutions gives rise to the idea and method employed in this work. Specifically, for better performance under a given LoRA rank setting, this paper proposes a fine-tuning strategy that promotes the growth of the intrinsic dimension of LoRA updates through regularization and gradient masking, bridging the gap between practical performance and theoretical optimal performance. The superiority of the two proposed techniques aligns with some of the previous studies. For example, the proposed regularizer encourages parameter space exploration. It helps reduce the performance gap as indicated by the performance benefits achieved by SoRA, which also tries to maintain a larger optimization space by keeping each epoch's sparsified components unchanged for the next epoch of updating (Ding et al., 2023). Additionally, the gradient masking method is shown to further improve generalization performance due to more efficient updates in certain LoRA directions. Such improvements can also be observed in DyLoRA, which only updates one row and column in LoRA matrices in each step (Valipour et al., 2023).

## 3 Preliminary

**Transformer Models.** A transformer-based pre-trained model typically involves $L$ stacked encoder/decoder blocks, with a multi-head attention module followed by a fully connected feed-forward network (FFN) in each block. Given an input sequence $\mathbf{X} \in \mathbb{R}^{n \times d}$, the output of the multi-head attention module can be written as:

$$
\begin{aligned}
&\text{MultiHead}(\mathbf{X}) = \text{Concat}(\text{head}_1, \dots, \text{head}_h)\mathbf{W}^O, \\
&\text{with head}_i = \text{Attention}(\mathbf{Q}_i, \mathbf{K}_i, \mathbf{V}_i), \\
&\text{and Attention}(\mathbf{Q}_i, \mathbf{K}_i, \mathbf{V}_i) = \text{softmax}\left(\frac{\mathbf{Q}_i \mathbf{K}_i^T}{\sqrt{d_k}}\right)\mathbf{V}_i,
\end{aligned}
\tag{1}
$$

where $\mathbf{Q}_i = \mathbf{X}\mathbf{W}_i^Q$, $\mathbf{K}_i = \mathbf{X}\mathbf{W}_i^K$, and $\mathbf{V}_i = \mathbf{X}\mathbf{W}_i^V$ are matrices of queries, keys, and values of $\text{head}_i$ respectively, with projection matrices $\mathbf{W}_i^Q \in \mathbb{R}^{d \times d_k}$, $\mathbf{W}_i^K \in \mathbb{R}^{d \times d_k}$, $\mathbf{W}_i^V \in \mathbb{R}^{d \times d_v}$, and $\mathbf{W}^O \in \mathbb{R}^{hd_v \times d}$. We refer readers to the original paper for a more comprehensive introduction to attention calculations in general (Vaswani et al., 2017).

Given the output of the multi-head attention module, the FFN further projects the $d$-dimensional output $\mathbf{X}'$ for each position. A two-layer FFN with a ReLU activation operates as follows:

$$
\text{FFN}(\mathbf{X}') = \max(0, \mathbf{X}'\mathbf{W}_{f_1} + b_1)\mathbf{W}_{f_2} + b_2,
\tag{2}
$$

where $\mathbf{W}_{f_1} \in \mathbb{R}^{d \times d_m}$ and $\mathbf{W}_{f_2} \in \mathbb{R}^{d_m \times d}$. Moreover, a residual connection followed by layer normalization is applied to each layer to generate the output of each transformer block given the input sequence $\mathbf{X}$ in the following way:

$$
\text{LayerNorm}(\mathbf{X} + \text{FFN}(\text{LayerNorm}(\mathbf{X} + \text{MultiHead}(\mathbf{X})))).
\tag{3}
$$

**LoRA Fine-tuning.** For a pre-trained matrix $\mathbf{W}_0 \in \mathbb{R}^{d_1 \times d_2}$, LoRA, as proposed by Hu et al. (2021), approximates its update $\Delta \mathbf{W}$ by $\Delta \mathbf{W} = \mathbf{BA}$, where $\mathbf{A} \in \mathbb{R}^{R \times d_2}$ and $\mathbf{B} \in \mathbb{R}^{d_1 \times R}$ with rank $R \ll \min(d_1, d_2)$ . During model fine-tuning, the weight matrix $\mathbf{W}_0$ is frozen, with only the LoRA adapters $\mathbf{A}$ and $\mathbf{B}$ being trainable. The modified LoRA forward pass is:

$$\mathbf{h} = \mathbf{W}_0\mathbf{x} + \Delta\mathbf{W}\mathbf{x} = \mathbf{W}_0\mathbf{x} + \mathbf{BA}\mathbf{x}. \tag{4}$$

Typically, the low-rank matrix $\mathbf{A}$ is initialized using a random Gaussian distribution, while $\mathbf{B}$ is initialized to zero, ensuring $\Delta\mathbf{W} = 0$ at the start of fine-tuning. Current approaches to fine-tuning large pre-trained models with LoRA apply a pair of matrices to all weight matrices involved in each transformer block's multi-head attention module and FFN (He et al., 2021; Zhang et al., 2023a).

**The Expressive Power of LoRA.** Zeng & Lee (2023) investigates the LoRA approximation error under a mild non-singularity assumption. To begin with, a $L$-layer width-$D$ fully connected ReLU neural network is denoted as $\mathrm{FNN}_{L,D}(\cdot; (\mathbf{W}_l)_{l=1}^L, (\mathbf{b}_l)_{l=1}^L)$, where $\mathbf{W}_l \in \mathbb{R}^{D \times D}$ are the weight matrices and $\mathbf{b}_l \in \mathbb{R}^D$ are the biases for each layer $l \in [L]$. In LoRA method, the primary objective is to adapt a pre-trained frozen FNN $f_0$ to approximate a target FNN $\bar{f}$, which are represented as follows:

$$\text{Target FNN } \bar{f} := \mathrm{FNN}_{\bar{L},D}(\cdot; (\bar{\mathbf{W}}_l)_{l=1}^L, (\bar{\mathbf{b}}_l)_{l=1}^L), \tag{5}$$

$$\text{Frozen FNN } f_0 := \mathrm{FNN}_{L,D}(\cdot; (\mathbf{W}_l)_{l=1}^L, (\mathbf{b}_l)_{l=1}^L), \tag{6}$$

where $\bar{\mathbf{W}}_l \in \mathbb{R}^{D \times D}$ and $\bar{\mathbf{b}}_l \in \mathbb{R}^D$ are the weight matrix and bias vector for the $l$-th layer of the target model $\bar{f}$, while $\mathbf{W}_l \in \mathbb{R}^{D \times D}$ and $\mathbf{b}_l \in \mathbb{R}^D$ are those for the $l$-th layer of the pre-trained frozen model $f_0$.

With a LoRA rank setting $R \in [D]$, the frozen FNN $f_0$ is adapted into a new model $f$:

$$\text{Adapted FNN } f := \mathrm{FNN}_{L,D}(\cdot; (\mathbf{W}_l + \Delta\mathbf{W}_l)_{l=1}^L, (\hat{\mathbf{b}}_l)_{l=1}^L), \tag{7}$$

where $\Delta\mathbf{W}_l \in \mathbb{R}^{D \times D}$ is the weight update approximated by LoRA with $\mathrm{rank}(\Delta\mathbf{W}_l) \leq R_l$, and $\hat{\mathbf{b}}_l$ is the updated bias vector or $l \in [L]$. Given that a large pre-trained model tends to be overparameterized, it is reasonable to assume that $L \geq \bar{L}$, which means the pre-trained model is much deeper than the target model to be approximated. Therefore, Zeng & Lee (2023) further introduces an ordered partition $\mathcal{P} = \{\mathcal{P}_1, \ldots, \mathcal{P}_{\bar{L}}\}$ to partition the $L$ layers in the adapted model $f$, such that $\bigcup_{i=1}^{\bar{L}} \mathcal{P}_i = [L]$. Each partition element $\mathcal{P}_i \in \mathcal{P}$ consists of consecutive integers $l \in \mathcal{P}_i$, which indicate the index of layers in the adapted model that will be used to approximate the $i$-th layer in the target model.

With the above definition, the following theoretical result provides an upper bound on the approximation error for the adapted model.

**Theorem 3.1 (Theorem 6 from (Zeng & Lee, 2023))** *If  $\sum_{l \in \mathcal{P}_i} R_l \geq rank(\bar{\mathbf{W}}_i - \prod_{l \in \mathcal{P}_i} \mathbf{W}_l)$ for all $i \in [\hat{L}]$, there exists LoRA adapters $(\Delta\mathbf{W}_l)_{l=1}^L$ with $rank(\Delta\mathbf{W}_l) \leq R_l$ and biases $(\hat{\mathbf{b}}_l)_{l=1}^L$ such that the adapted model $f$ can exactly approximate the target model $\hat{f}$.*

*Furthermore, define the approximation error of the $i$-th layer as $e_i = \sigma_{\sum_{l \in \mathcal{P}_i} R_l + 1}(\bar{\mathbf{W}}_i - \prod_{l \in \mathcal{P}_i} \mathbf{W}_l)$, and the magnitude of the weight parameters and the input as*

$$\beta := \max_{i \in [\bar{L}]} \left( \sqrt{\|\mathbf{\Sigma}\|_F} \prod_{j=1}^i \|\bar{\mathbf{W}}_j\|_F + \sum_{j=1}^i \prod_{k=j+1}^{i-1} \|\bar{\mathbf{W}}_k\|_F \|\bar{\mathbf{b}}_j\|_2 \right) \vee \sqrt{\|\mathbf{\Sigma}\|_F}.$$

*Then, there exists LoRA adapters $(\Delta\mathbf{W}_l)_{l=1}^L$ with $rank(\Delta\mathbf{W}_l) \leq R_l$ and biases $(\hat{\mathbf{b}}_l)_{l=1}^L$ such that for any input $\mathbf{x}$ with $\mathbb{E}\mathbf{x}\mathbf{x}^T = \mathbf{\Sigma}$, the approximation error can be bounded as*

$$\mathbb{E}\left\|f(\mathbf{x}) - \bar{f}(\mathbf{x})\right\|_2 \leq \beta \sum_{i=1}^{\bar{L}} \max_{k \in [\bar{L}]} \left(\|\bar{\mathbf{W}}_k\|_F + e_k\right)^{\bar{L}-i} e_i. \tag{8}$$

In the above bound, $\beta$ and $\|\bar{\mathbf{W}}_k\|_F$ capture the magnitude of the weight parameters in the target model and the input. The LoRA rank setting $R_l$ for all layers $l \in [L]$ in the adapted model contributes to this bound through the term $e_i$ for all $i \in [\bar{L}]$. The following section focuses on the interconnection among the constituting parts of the $e_i$ term and explains how this theoretical conclusion can be utilized to enhance LoRA adaptation on real-world datasets.

## 4 RM-LoRA Method

Based on the analysis of LoRA's expressive power described in Section 3, we introduce **R**egularized and **M**asked LoRA (RM-LoRA), a robust method aimed at enhancing the generalization performance of LoRA within a given parameter budget. This section presents the details of RM-LoRA, including its underlying principles and specific techniques.

### 4.1 Influence of the Intrinsic Dimension of LoRA Adapter $\Delta \mathbf{W}$

Note that the partition $\mathcal{P}_i$ of the pre-trained model for the $i$-th layer in the target model is an intrinsic but unknown property during adaptation, and consequently, the number and index of layers $l \in \mathcal{P}_i$ are also unknown. Nevertheless, with a pre-trained model $f_0$ to be adapted and the target model $\bar{f}$ determined by a given downstream task, the partition can be considered deterministic, as can the discrepancy between the pre-trained model and the target model $\mathbf{E}_i = \bar{\mathbf{W}}_i - \prod_{l \in \mathcal{P}_i} \mathbf{W}_l$. Consider the case where the LoRA rank setting for each layer $l \in \mathcal{P}_i$ is the same as $R$. The $e_i$ term in Equation 8 can be rewritten as:

$$e_{i,\mathrm{rank}(\Delta \mathbf{W}_{l \in \mathcal{P}_i}) \leq R} = \sigma_{\sum_{l \in \mathcal{P}_i} R + 1}(\mathbf{E}_i) \; \textit{for each layer } i \in [\bar{L}], \tag{9}$$

where $\mathrm{rank}(\Delta \mathbf{W}_{l \in \mathcal{P}_i}) \leq R$ represents the LoRA adapter for each layer $l \in \mathcal{P}_i$ satisfies the rank constraint $\mathrm{rank}(\Delta \mathbf{W}_l) \leq R$.

Clearly, increasing $\mathrm{rank}(\Delta \mathbf{W}_l)$ helps relax the constraint on LoRA rank $R$ to achieve a certain level of approximation error. For example, consider two LoRA rank settings $R_1$ and $R_2$ with $R_1 < R_2$. If $\mathrm{rank}(\Delta \mathbf{W}_l) \leq R_1 < R_2$, then $e_{i,\mathrm{rank}(\Delta \mathbf{W}_{l \in \mathcal{P}_i}) \leq R_2}$ degenerates to $e_{i,\mathrm{rank}(\Delta \mathbf{W}_{l \in \mathcal{P}_i}) \leq R_1}$ for $i \in [\bar{L}]$, despite the larger size of LoRA matrices under the LoRA setting of $R_2$. In this case, LoRA rank $R_1$ and $R_2$ yield the same LoRA approximation error $\mathbb{E}\|f(\mathbf{x}) - \bar{f}(\mathbf{x})\|_2$ according to Equation 8.

### 4.2 Regularization on LoRA Weights

Let a pair of LoRA low-rank matrices be denoted as $\mathbf{W}^A$ and $\mathbf{W}^B$, respectively. To enforce the growth in rank of $\Delta \mathbf{W} = \mathbf{W}^B \mathbf{W}^A$, the following regularizer is first used to encourage $\mathbf{W}^A$ and $\mathbf{W}^B$ to be orthogonal:

$$\mathrm{Reg}(\mathbf{W}^A, \mathbf{W}^B) = \|\mathbf{W}^A(\mathbf{W}^A)^\top - \mathbf{I}\|_F^2 + \|(\mathbf{W}^B)^\top \mathbf{W}^B - \mathbf{I}\|_F^2. \tag{10}$$

The orthogonality of $\mathbf{W}^A$ and $\mathbf{W}^B$ helps increase the $\mathrm{rank}(\mathbf{W}^A)$ and $\mathrm{rank}(\mathbf{W}^B)$. According to the lower bound for the rank of the matrix product, for matrices $\mathbf{A} \in \mathbb{R}^{R \times d_2}$ and $\mathbf{B} \in \mathbb{R}^{d_1 \times R}$, the rank of their product matrix $\mathbf{C} = \mathbf{BA}$ satisfies $\mathrm{rank}(\mathbf{C}) \geq \max(\mathrm{rank}(\mathbf{A}) + \mathrm{rank}(\mathbf{B}) - R, 0)$. This lower bound ensures the growth of the intrinsic rank of the LoRA adapter $\Delta \mathbf{W} = \mathbf{W}^A \mathbf{W}^B$ as the $\mathrm{rank}(\mathbf{W}^A)$ and $\mathrm{rank}(\mathbf{W}^B)$ increase with the regularizer shown in Equation 10. Note that there exist other alternative regularizers that theoretically can also encourage the growth of $\mathrm{rank}(\Delta \mathbf{W})$, but are infeasible in reality due to considerations of differentiability, numerical stability, and computational costs[1].

### 4.3 Gradient Masking for Partial Updates

The gradient masking algorithm in RM-LoRA is designed to perform partial updates in LoRA matrices. The algorithm takes as input the total number of steps $T$, the LoRA rank $R$, and the number of directions $\hat{r}$ to

---

[1]The nuclear (trace) norm involves expensive computation for singular value decomposition, especially with large matrices. Regularization on the determinants suffers from numerical instability since the determinant calculation is highly sensitive to small changes in the matrix elements. Constraints on eigenvalues or singular values of the matrix are not directly differentiable.

---

**Algorithm 1** Gradient Masking Algorithm

---

1: **Input:** Total steps $T$, LoRA rank $R$, number of updated directions $\hat{r}$.
2: **for** $t = 0$ to $T - 1$ **do**
3:     **for** each pair of LoRA weight matrices $(\mathbf{W}_t^A, \mathbf{W}_t^B)$ in the model **do**
4:         Sample a mini-batch data $\xi_t$ and compute the gradients $(\nabla_{\xi_t} \mathbf{W}_t^A, \nabla_{\xi_t} \mathbf{W}_t^B)$;
5:         Initialize gradient masks $(\mathbf{M}_t^A, \mathbf{M}_t^B) \leftarrow \mathbf{0}$ with the same shape as $(\nabla_{\xi_t} \mathbf{W}_t^A, \nabla_{\xi_t} \mathbf{W}_t^B)$;
6:         Construct the set $\mathcal{R}_t$ by randomly selecting $\hat{r}$ distinct integers from $\{1, 2, \ldots, R\}$.
7:         **for** each $i$ in $\mathcal{R}_t$ **do**
8:           $\mathbf{M}_t^A[i, j] = 1$ for all $j = 1, 2, \ldots, R$.
9:         **end for**
10:        **for** each $j$ in $\mathcal{R}_t$ **do**
11:          $\mathbf{M}_t^B[i, j] = 1$ for all $i = 1, 2, \ldots, R$.
12:        **end for**
13:        Apply gradient mask $\nabla_{\xi_t} \mathbf{W}_t^A \leftarrow \nabla_{\xi_t} \mathbf{W}_t^A \odot \mathbf{M}_t^A$, $\nabla_{\xi_t} \mathbf{W}_t^B \leftarrow \nabla_{\xi_t} \mathbf{W}_t^B \odot \mathbf{M}_t^B$.
14:        Perform optimization step $\mathbf{W}_t^A = \mathbf{W}_t^A - \eta \nabla_{\xi_t} \mathbf{W}_t^A$, $\mathbf{W}_t^B = \mathbf{W}_t^B - \eta \nabla_{\xi_t} \mathbf{W}_t^B$.
15:     **end for**
16: **end for**
17: **Output:** Updated LoRA weight matrices $(\mathbf{W}_T^A, \mathbf{W}_T^B)$ for each fine-tuned module.

---

update in each step. In each training step $t$, it samples a mini-batch of data $\xi_t$ and computes the gradients $\nabla_{\xi_t} \mathbf{W}_t^A$ and $\nabla_{\xi_t} \mathbf{W}_t^B$ for each pair of LoRA weight matrices. The corresponding gradient masks are first initialized to zero, before a set $\mathcal{R}_t$ of $\hat{r}$ distinct directions is randomly selected. The RM-LoRA method then sets the relevant entries in the gradient masks to one according to the selected directions. These masks are applied to the gradients to restrict the update directions. Finally, the algorithm updates the weight matrices $\mathbf{W}_t^A$ and $\mathbf{W}_t^B$ using the masked gradients, thus achieving the partial update of LoRA weight matrices. The complete process of gradient masking is summarized in Algorithm 1.

By strategically updating a subset of directions, gradient masking helps promote sparsity in gradient flow, which aligns with the inherent low-rank structure of many real-world datasets. To further improve efficiency, the gradient masking process can be optimized by avoiding the computation of gradients for the masked positions altogether. By skipping these unnecessary calculations, the computational overhead involved in the backpropagation phase can be significantly reduced, especially in large-scale models with high-dimensional parameter spaces. Modern training frameworks, such as PyTorch, provide native support for mechanisms like partial gradient computation, which can be leveraged to achieve a more efficient implementation of the proposed gradient masking.

## 5 Experiments

With the regularization and gradient masking technique as described in Section 4, the proposed RM-LoRA method is expected to alleviate the problem of overfitting and achieve enhanced generalization for fine-tuning pre-trained models. In this section, RM-LoRA is comprehensively evaluated against state-of-the-art LoRA variants on multiple datasets.

### 5.1 Experimental Setup

The details of the experiment for the evaluation of the proposed RM-LoRA method are outlined as follows:

**Models and Datasets**. This paper compares our proposed RM-LoRA with the original LoRA and its recent variants across both computer vision and natural language tasks. For the vision task, we fine-tune a Vision Transformer (ViT-B/16) model, which has 86M parameters. This model was pre-trained on ImageNet dataset as described in the original paper (Dosovitskiy et al., 2020). Fine-tuning is performed on the CIFAR-100 dataset and Food-101 dataset to evaluate our approach in the computer vision domain. For language tasks, we fine-tune the DeBERTa-v3-base model with 184M parameters (He et al., 2022), and a GPT-2 Small model

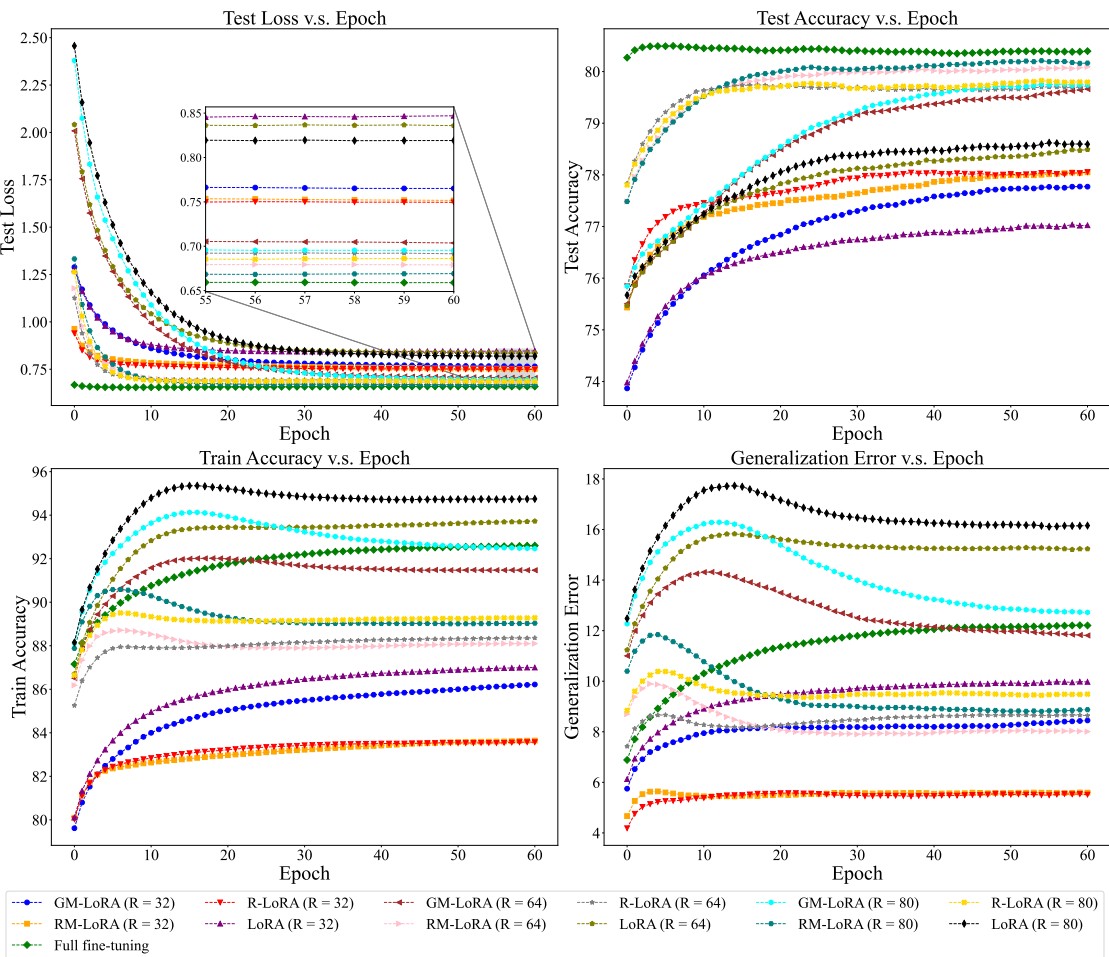

Figure 1: Results with ViT model on CIFAR-100.

with 117M parameters. Evaluations are conducted on the General Language Understanding Evaluation (GLUE) benchmark (Wang et al., 2019) for language understanding and the Stanford Question Answering Dataset (SQuAD 1.1) (Rajpurkar et al., 2016) for question answering.

**Baselines**. The following baselines are implemented within the same HuggingFace's Transformers framework for fair comparison (Wolf et al., 2019):

- *Full fine-tuning* (FT) uses the pre-trained model as the initialization point and updates all parameters in the model through gradient backpropagation, resulting in a very large trainable parameter budget that may be impractical in resource-limited environments.

- *LoRA* (Hu et al., 2021) approximates the incremental updates in pre-trained model weights by using the product of two trainable matrices with rank $R$, which significantly reduces the number of trainable parameters needed to adapt pre-trained models while achieving competitive generalization performance on downstream tasks.

- *AdaLoRA* (Zhang et al., 2023a) uses the product of three small matrices in the form of singular value decomposition to parameterize the updates in pre-trained model weights, and then prunes the singular values of lower importance in the diagonal matrix to achieve a pre-set total parameter budget $b$ across all adapter weight matrices. Compared to LoRA with a fixed parameter budget, AdaLoRA provides a more flexible solution for task-specific fine-tuning, resulting in a better efficiency-generalization trade-off.

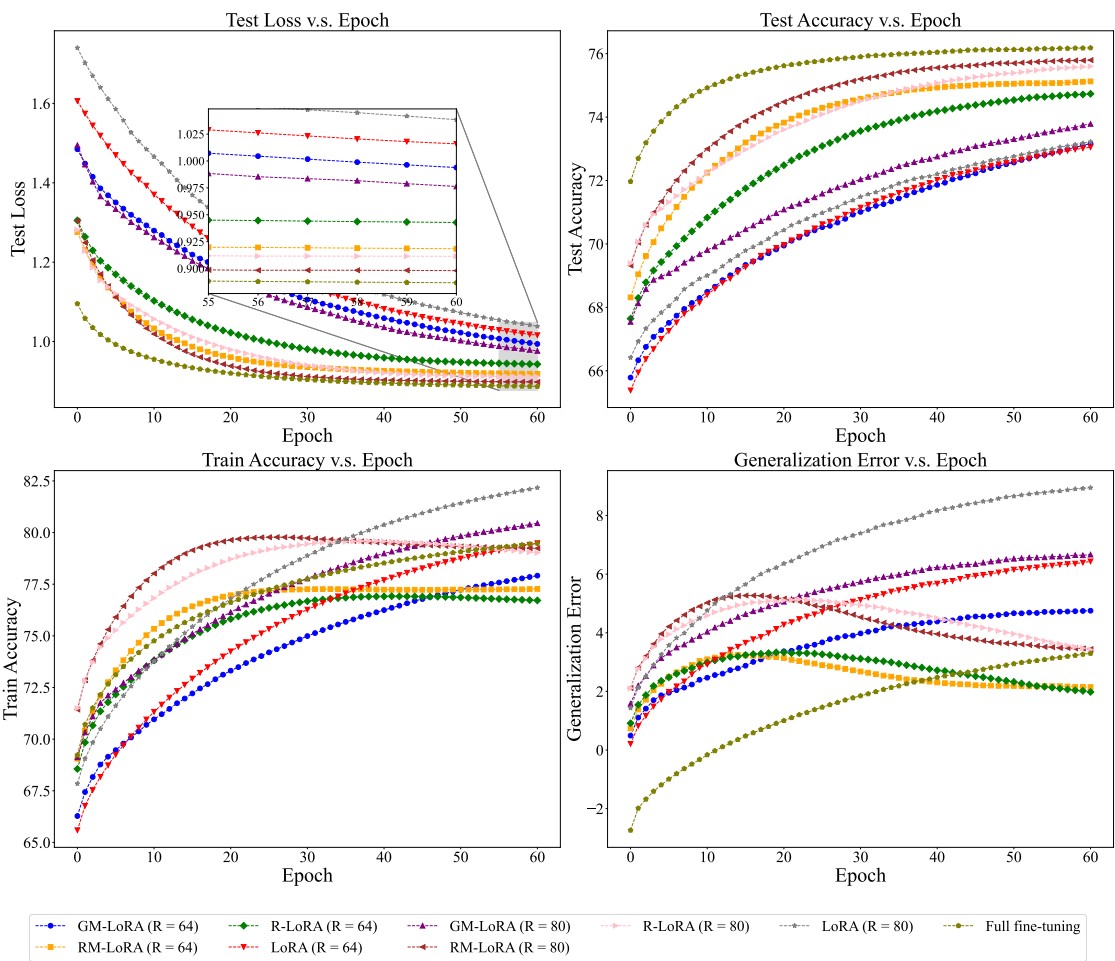

Figure 2: Results with ViT model on Food-101.

- *SoRA* (Ding et al., 2023) parameterizes the updates in pre-trained model weights similarly to AdaLoRA, with an additional gate unit in between, and controls the sparsity of the gate by pruning components with absolute values lower than a pre-set threshold $\lambda$. By retaining pruned components until the pruning step in the final epoch, SoRA maintains a larger parameter space for exploration during fine-tuning compared to AdaLoRA.

These aforementioned LoRA methods aim to strike a balance between parameter efficiency and generalization performance in adapting pre-trained models. Sharing this goal, our proposed RM-LoRA addresses the challenge by exploring and enhancing the role of the intrinsic dimensions of LoRA matrices. For a fair comparison of all the LoRA variants, including our proposed RM-LoRA method, their performance is evaluated under the **same parameter budget during inference**. All experiments are conducted across three independent runs with different random seeds, as discussed in the following sections of this paper.

## 5.2 Image Classification

Figure 1 illustrates the results achieved by the ViT model on the CIFAR-100 dataset, serving as a preliminary measure of the performance of LoRA and the enhancement techniques for LoRA method proposed in this paper. To simulate the theoretical results based on the fully connected layer, only the **last classification layer** of the ViT model is fine-tuned by LoRA low-rank matrices. The four sub-figures of Figure 1 display

Table 1: Results with DeBERTaV3 model on GLUE benchmark

| Method | # T / I - Params | CoLA Mcc | STS-B Pearson / Spearman | QNLI Acc | MNLI Acc | WNLI Acc | RTE Acc | MRPC Acc / F1 | QQP Acc / F1 | SST-2 Acc |
|---|---|---|---|---|---|---|---|---|---|---|
| Full FT | 184M / 184M | 0.668 | 0.892 / 0.890 | 0.940 | 0.907 | 0.563 | 0.852 | 0.853 / 0.891 | 0.918 / 0.891 | 0.934 |
| LoRA$_{R=4}$ | 1.26M / 1.26M | 0.680 | 0.912 / 0.912 | 0.939 | 0.901 | 0.718 | 0.856 | 0.892 / 0.922 | 0.912 / 0.883 | 0.934 |
| AdaLoRA | 1.92M / 1.26M | 0.663 | 0.905 / 0.909 | 0.934 | 0.904 | 0.563 | 0.845 | 0.880 / 0.911 | 0.912 / 0.884 | **0.955** |
| SoRA | 1.92M / 1.38M | 0.655 | 0.905 / 0.907 | 0.926 | 0.889 | 0.690 | 0.848 | 0.895 / 0.924 | 0.820 / 0.762 | 0.820 |
| R-LoRA$_{R=4}$ | 1.26M / 1.26M | 0.685 | 0.914 / 0.914 | 0.941 | 0.903 | 0.718 | 0.863 | **0.897 / 0.925** | **0.917 / 0.889** | 0.943 |
| RM-LoRA$_{R=4}$ | 1.26M / 1.26M | **0.689** | **0.915 / 0.915** | **0.942** | **0.906** | **0.732** | **0.866** | 0.892 / 0.922 | 0.913 / 0.884 | 0.943 |
| LoRA$_{R=8}$ | 1.92M / 1.92M | 0.672 | 0.913 / 0.914 | 0.938 | 0.900 | 0.718 | 0.856 | 0.900 / 0.928 | 0.916 / 0.889 | 0.936 |
| AdaLoRA | 3.25M / 1.92M | 0.664 | 0.912 / 0.913 | 0.942 | 0.906 | 0.578 | 0.841 | 0.897 / 0.924 | 0.912 / 0.885 | 0.948 |
| SoRA | 3.25M / 2.24M | 0.665 | 0.907 / 0.910 | 0.930 | 0.895 | 0.690 | 0.841 | 0.897 / 0.927 | 0.819 / 0.761 | 0.826 |
| R-LoRA$_{R=8}$ | 1.92M / 1.92M | **0.694** | 0.914 / 0.914 | 0.943 | 0.904 | 0.732 | 0.863 | 0.907 / 0.933 | **0.917 / 0.890** | 0.940 |
| RM-LoRA$_{R=8}$ | 1.92M / 1.92M | 0.688 | **0.915 / 0.915** | **0.944** | **0.907** | **0.747** | **0.870** | **0.914 / 0.938** | 0.913 / 0.885 | **0.948** |

Table 2: Results with DeBERTaV3 model on SQuAD

| Method | # T / I - Rank | | # T / I - Params | | EM | F1 Score |
|---|---|---|---|---|---|---|
| Full FT | N/A | N/A | **184M** | **184M** | 87.83 ± 0.01 | 93.62 ± 0.01 |
| LoRA$_{R=8}$ | 8 | 8 | 1.33M | 1.33M | 87.47 ± 0.01 | 93.33 ± 0.01 |
| SoRA$_{\lambda_2=5e-4}$ | 16 | 16 | 2.66M | 1.28M | 82.73 ± 0.02 | 90.49 ± 0.01 |
| **RM-LoRA$_{R=8}$** | 8 | 8 | **1.33M** | **1.33M** | **87.87 ± 0.03** | **93.67 ± 0.01** |
| LoRA$_{R=4}$ | 4 | 4 | 0.67M | 0.67M | 87.43 ± 0.01 | 93.35 ± 0.02 |
| AdaLoRA | 8 | 4 | 1.33M | 0.67M | 86.48 ± 0.02 | 92.83 ± 0.01 |
| SoRA$_{\lambda_2=5e-4}$ | 8 | 8 | 1.33M | 0.61M | 79.64 ± 0.02 | 88.01 ± 0.02 |
| **RM-LoRA$_{R=4}$** | 4 | 4 | **0.67M** | **0.67M** | **87.57 ± 0.03** | **93.51 ± 0.02** |

the test loss, test accuracy, train accuracy, and generalization error (measured by train accuracy minus test accuracy) for each method respectively.

As observed in Figure 1, the regularization and gradient masking techniques proposed in this paper both effectively mitigate overfitting and achieve higher accuracy on the test dataset. Specifically, Regularized LoRA (**R-LoRA**) and Gradient Masking LoRA (**GM-LoRA**) represent the application of each technique individually, while RM-LoRA combines them together as described in Section 4. Furthermore, Table 5 presents the orthogonal loss $\|\Delta \mathbf{W}(\Delta \mathbf{W})^\top - \mathbf{I}\|_F^2$ of $\Delta \mathbf{W}$ after being fine-tuned by each method, which describes its spatial distribution. The results in Table 5 demonstrate that the orthogonal penalty term for the LoRA matrices $\mathbf{W}_A$ and $\mathbf{W}_B$ in Eq. 10 effectively promotes the orthogonality of their product. Meanwhile, as the orthogonal loss of $\Delta \mathbf{W}$ decreases, the accuracy of LoRA fine-tuning on the test data increases. Therefore, by effectively promoting the reduction of $\Delta \mathbf{W}$'s orthogonal loss, the RM-LoRA method proposed in this paper achieves the best generalization performance across all rank settings.

Figure 2 presents the results achieved by the ViT model on the Food-101 dataset, which exhibit similar trends to those observed in Figure 1 and further demonstrate the generalizability of the proposed advancements.

## 5.3  Natural Language Understanding

The GLUE benchmark includes two single-sentence classification tasks (CoLA, SST-2), three similarity and paraphrase tasks (MRPC, STS-B, QQP), and four natural language inference tasks (QNLI, WNLI, MNLI, RTE). The proposed RM-LoRA method is compared against the baseline methods under multiple LoRA rank settings to demonstrate its superiority. Table 1 shows the performance achieved by different methods on GLUE tasks, as well as the number of trainable and inference parameters (*# T / I - Params* respectively). The standard deviations (STD) across three independent runs are limited within 0.01 but are omitted in the table due to space constraints. The best result for each task is highlighted in **bold**. R-LoRA with the proposed regularizer consistently achieves performance gains under the same or lower inference parameter budget compared to other methods in most cases. Furthermore, RM-LoRA with gradient masking

Table 3: Results with GPT-2 model on SQuAD

| Method | # T / I - Rank | | # T / I - Params | | EM | F1 Score |
|---|---|---|---|---|---|---|
| Full FT | N/A | N/A | **117M** | **117M** | **63.30 ± 0.01** | **74.41 ± 0.02** |
| LoRA$_{R=16}$ | 16 | 16 | 1.69M | 1.69M | 61.41 ± 0.02 | 72.76 ± 0.03 |
| AdaLoRA | 32 | 16 | 3.39M | 1.69M | 61.70 ± 0.01 | 72.30 ± 0.03 |
| SoRA$_{\lambda_2=5e-4}$ | 32 | 32 | 3.39M | 1.62M | 60.80 ± 0.02 | 71.80 ± 0.04 |
| **RM-LoRA**$_{R=16}$ | 16 | 16 | **1.69M** | **1.69M** | **62.80 ± 0.01** | **73.50 ± 0.02** |
| LoRA$_{R=8}$ | 8 | 8 | 0.85M | 0.85M | 60.34 ± 0.02 | 71.92 ± 0.02 |
| AdaLoRA | 16 | 8 | 1.69M | 0.85M | 60.20 ± 0.02 | 71.40 ± 0.02 |
| SoRA$_{\lambda_2=5e-4}$ | 16 | 16 | 1.69M | 0.81M | 59.80 ± 0.02 | 70.90 ± 0.03 |
| **RM-LoRA**$_{R=8}$ | 8 | 8 | **0.85M** | **0.85M** | **61.00 ± 0.01** | **72.20 ± 0.01** |

Table 4: Hyperparameters for GLUE tasks

| Datasets | Learning Rate | | Batch Size | # Epochs | SoRA Parameters | |
|---|---|---|---|---|---|---|
| | FT | LoRA | | | $\lambda$ | Sparsity |
| CoLA | 0.00001 | 0.001 | 32 | 3 | 0.001 | 68.96% |
| SST-2 | 0.00001 | 0.001 | 32 | 3 | 0.005 | 53.90% |
| MRPC | 0.00001 | 0.001 | 32 | 5 | 0.001 | 79.27% |
| STS-B | 0.00001 | 0.001 | 32 | 3 | 0.001 | 80.62% |
| QQP | 0.00001 | 0.001 | 32 | 3 | 0.005 | 53.90% |
| QNLI | 0.00001 | 0.001 | 32 | 3 | 0.0001 | 58.06% |
| MNLI | 0.00001 | 0.001 | 32 | 3 | 0.005 | 64.77% |
| WNLI | 0.00001 | 0.001 | 32 | 5 | 0.01 | 62.53% |
| RTE | 0.00001 | 0.001 | 32 | 3 | 0.005 | 64.23% |

outperforms R-LoRA in a majority of task settings. The specific fine-tuning hyperparameters adopted by each method on the GLUE benchmark are summarized in Table 4.

Table 5: Correlation between the orthogonality of $\mathbf{\Delta W}$ and generalization performance

| Method | Rank of $\Delta W$ | Orthogonality Loss of $\Delta W$ | Test Acc (mean ± std) |
|---|---|---|---|
| LoRA$_{R=32}$ | 29 | 585.835 | 76.97 ± 0.02 |
| GM-LoRA$_{R=32}$ | 29 | 277.919 | 77.72 ± 0.02 |
| R-LoRA$_{R=32}$ | 30 | 122.079 | 77.99 ± 0.03 |
| **RM-LoRA**$_{R=32}$ | 30 | **95.240** | **78.05 ± 0.02** |
| LoRA$_{R=64}$ | 55 | 410.154 | 78.41 ± 0.03 |
| GM-LoRA$_{R=64}$ | 52 | 175.010 | 79.60 ± 0.05 |
| R-LoRA$_{R=64}$ | 59 | 79.876 | 79.66 ± 0.06 |
| **RM-LoRA**$_{R=64}$ | 57 | **58.010** | **80.07 ± 0.04** |
| LoRA$_{R=80}$ | 64 | 342.197 | 78.13 ± 0.03 |
| GM-LoRA$_{R=80}$ | 59 | 155.010 | 79.68 ± 0.02 |
| R-LoRA$_{R=80}$ | 72 | 68.970 | 79.63 ± 0.05 |
| **RM-LoRA**$_{R=80}$ | 69 | **48.832** | **80.20 ± 0.02** |

### 5.4 Question Answering

The performance of different methods using the DeBERTaV3 model and the GPT-2 model on the SQuAD dataset is shown in Table 2 and Table 3. Similarly, the proposed RM-LoRA method outperforms other baselines under the same or lower inference parameter budget across varying LoRA rank settings, with EM denoting the average exact match score and F1 referring to the average F1 score. These results highlight that the RM-LoRA method consistently improves LoRA's fine-tuning performance across various benchmarks. The capability of achieving better or comparable performance with a reduced parameter budget is especially significant for practical deployment in mobile systems, where efficiency and resource utilization are crucial factors.

## 6 Conclusion

In conclusion, the exploration of the intrinsic dimension in LoRA fine-tuning reveals critical insights into optimizing parameter efficiency and enhancing model generalization. The theoretical foundation indicates that the intrinsic dimension of the approximated matrix updates is more pivotal in achieving effective LoRA fine-tuning than the previously emphasized LoRA rank. By employing a regularization technique and a gradient masking method to encourage parameter space exploration while controlling the trainable parameters budget, this paper presents an advanced low-rank adaptation strategy that addresses the challenges of sub-optimal performance associated with LoRA. The better generalization performance achieved by the proposed RM-LoRA under the same or lower parameter budget compared to other methods represents progress in the field of parameter-efficient fine-tuning for large pre-trained models.

## 7 Acknowledgement

We would like to acknowledge the support from the National Key R&D Program of China under Grant No. 2022ZD0160504 and the Shenzhen Key Laboratory of Ubiquitous Data Enabling under Grant No. ZDSYS20220527171406015. Their support has been invaluable in the completion of this work.

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
