# OpenReview forum: "Enhancing Parameter Efficiency and Generalization in Large Models: A Regularized and Masked Low-Rank Adaptation Approach"
_TMLR — Accepted by TMLR_

### Review · Reviewer_u7Bu · 2024-11-20

**Summary Of Contributions:**

This paper focuses on improving PEFT, specifically the performance of LoRA. With the goal of increasing the intrinsic rank of the LoRA matrix $\Delta W$, the paper proposes adding a regularizer on the LoRA matrices to make the rows/columns orthogonal to each other. Additionally, the paper proposed using masked gradient updates to randomly update $\hat{r}$ rows/columns in the LoRA matrices.
Numerical results are conducted on CIFAR-100 with ViT and GLUE&SQuAD with DeBERTa-V3. In the experiments, the paper compares the proposed method (R-LoRA, GM-LoRA, and RM-LoRA) with several baselines, including full FT, LoRA, AdaLoRA, and SoRA. Numerical results show that the proposed RM-LoRA outperforms existing algorithms. The results also show that there is a strong correlation between the orthogonality and the FT model performance.

**Audience:**

Yes

**Claims And Evidence:**

No

**Requested Changes:**

Please address the above weaknesses, specifically:
1. Clarify the mismatch of the numbers in the experiments.
2. The STD of multiple runs
3. Clarify the models used
4. Add comparison on more models

Minor issues:
1. Change the color or line type in Fig. 1 to improve readability.
2. Add a discussion about the algorithms' connection

**Strengths And Weaknesses:**

Strength:
1. The motivation of the regularizer is clearly explained.
2. The paper provides numerical results to validate the proposed algorithms' motivation by Tab. 4.

Weakness:
1. The motivation of the gradient masking requires extra explanation. Sec. 3.4 is too short to explain the proposed gradient masking method.
2. Lacks discussion and comparison with existing methods. The description in Sec. 4.1 on the baselines should be moved to Sec. 3 with a more detailed discussion on the connection and difference between the proposed method and these baselines.
3. In sufficient numerical results. Since the paper is purely empirical, more substantial numerical evidence is required.
    1. The STD is missing in the tables. The experiments should report the STD of multiple runs so readers can decide if the performance improvement is statistically significant, especially when the algorithm introduces extra randomness in sampling $\hat{r}$.
    2. The lines in Fig. 1 are hard to distinguish; please consider using more distinguishable color/line type coding.
    3. The reported results for ViT do not match SOTA results, e.g., in [This GitHub Project](https://github.com/bwconrad/vit-finetune), the reported test acc. using LoRA goes to 92% for CIFAR-100, in [This Paper](https://arxiv.org/pdf/2406.16282), it is 92%. However, the paper only reports less than 81%. A similar issue appears in the results of AdaLoRA, where the reported numbers do not match the ones in the [original paper](https://arxiv.org/pdf/2303.10512).
    4. The ViT and DeBERTa-V3 model size is not specified. Since these models have many different settings, the author should provide the specific choice used for the experiment, including the size and the pretrained source.
    5. The numerical result is limited to only two models, ViT and DeBERTa-V3. The author should compare the performance of more models, e.g., OPT and Llama.

---

> ### Author Response · Authors · 2024-12-04
> **Response to Reviewer u7Bu's Comments**
>
> We sincerely thank the reviewer for the very constructive and careful comments on our paper. We try to address the concerns listed below.
>
> > Q1: The motivation for gradient masking requires extra explanation. Section 3.4 is too short to explain the method.
>
> Thank you for raising this concern. The motivation behind introducing gradient masking is to promote sparsity in gradient flow and introduce stochasticity into updates, which can help avoid overfitting by encouraging exploration of a broader parameter space. We have expanded the corresponding section (now Section 4.3) to provide a clearer explanation. We sincerely appreciate the reviewer’s constructive comment, which has helped us improve our manuscript.
> > Q2: The baseline discussion in Section 4.1 should be moved to Section 3 with more detailed connections and differences.
>
> Thank you for the suggestion. To maintain a comprehensive description of the experimental settings in Section 4, we have retained the baseline discussion in its original location. However, we have added explanations in this section to clarify the connections and differences among the baselines, elaborating on how our proposed method builds upon or diverges from these approaches. We hope this adjustment clarifies the connection to prior works and provides a stronger foundation for understanding our contributions.
>
> > Q3: STD of multiple runs is missing in the tables.
>
> Thank you for the comment. We have reported the STD information across three independent runs in Tables 2 and 3 of the revised manuscript. For Table 1, we included the STD information in the description section rather than directly in the table due to space constraints.
>
> > Q4: The reported results for ViT and AdaLoRA do not match prior benchmarks.
>
> Thank you for pointing this out. We would like to clarify the following:
>
> - **Vision Transformer (ViT)**:
>   As illustrated in Section 5.2 of our manuscript, we fine-tune only the last classification layer of the ViT model using LoRA low-rank matrices. This setup was chosen to simulate the theoretical results of LoRA, which were developed specifically for fully connected layers, as discussed in [1]. Consequently, our experimental setup differs from some prior benchmarks where additional layers may also be fine-tuned. We have made this distinction explicit in the revised manuscript to avoid confusion.
>
> - **AdaLoRA**:
>   For AdaLoRA, we attempted to reproduce the original experimental results using the publicly available codebase (https://github.com/QingruZhang/AdaLoRA). Additionally, we integrated our proposed advancements for LoRA into the AdaLoRA framework to ensure a fair comparison. The results reported in our manuscript reflect our findings under identical experimental settings for all methods. We are also currently organizing our code and will make it publicly available after the anonymous review process.
>
>
> > Q5: The ViT and DeBERTa-V3 model size is not specified.
>
> Thank you for raising this concern. We apologize for the oversight and provide the following details for clarity in the revised manuscript. We used the ViT-B/16 model, which has 86M parameters and was pre-trained on ImageNet as described in the original paper [2]. For DeBERTa-v3, we used the DeBERTa-v3-base model with 184M parameters [3].
>
> > Q6: Numerical results are limited to only ViT and DeBERTa-V3. Additional models should be included.
>
> Thank you for this suggestion. We selected these two models—ViT and DeBERTa-V3—because they represent two major domains, computer vision (CV) and natural language processing (NLP), respectively. Our goal was to use representative models from each field to provide a general validation of the generalizability of our proposed method.
>
> We agree that including additional models could further demonstrate the broader applicability of our approach. In our revised manuscript, we have added the results for the GPT-2 model in Table 3. However, due to resource constraints and the limited timeline for this rebuttal, we are unable to conduct more additional experiments to further strengthen our conclusions. We sincerely appreciate your valuable feedback and will include more results in a follow-up study or future publication.
>
> > Q7: The lines in Fig. 1 are hard to distinguish.
>
> We have updated Figure 1 to improve readability. We sincerely thank the reviewer for this valuable feedback.

---

> > ### Author Response · Authors · 2024-12-04
> > **References**
> >
> > References:
> >
> > *[1] Yuchen Zeng and Kangwook Lee. The Expressive Power of Low-Rank Adaptation. In International Conference on Learning Representations, 2023.*
> >
> > *[2] Alexey Dosovitskiy, Lucas Beyer, Alexander Kolesnikov, Dirk Weissenborn, Xiaohua Zhai, Thomas Unterthiner, Mostafa Dehghani, Matthias Minderer, Georg Heigold, Sylvain Gelly, et al. An image is worth 16x16 words: Transformers for image recognition at scale. In International Conference on Learning Representations, 2020.*
> >
> > *[3] Pengcheng He, Jianfeng Gao, and Weizhu Chen. DeBERTaV3: Improving DeBERTa using ELECTRA-Style Pre-Training with Gradient-Disentangled Embedding Sharing. In International Conference on Learning Representations, 2023.*

---

### Review · Reviewer_LdFz · 2024-11-22

**Summary Of Contributions:**

This paper propose improve LoRa by adding regularization and mask mechanisms. It did so by investigating the intrinsic dimension. It proposes that employing regularization and a gradient masking method could encourage higher intrinsic dimension. The proposed method achieves superior generalization performance with the same or lower trainable parameter budget.

**Audience:**

Yes

**Broader Impact Concerns:**

The reviewer does not see broader impact concerns with regard to this paper.

**Claims And Evidence:**

Yes

**Requested Changes:**

It would be better to not discuss other work's theorem in the main section.

**Strengths And Weaknesses:**

**Strength**

This paper addresses the fundamental problem of improving LoRA efficiency with the proposed regularization method.

If the algorithm is implemented robustly and scale to industry infra, it could help to large scale model training.

The experiments offers comprehensive comparison of various LoRA baselines, which helps readers to understand its effectiveness.

**Weakness**

The paper discussed theorems in the main section yet only to cite other's contribution. This should be better put to related works section.

---

> ### Author Response · Authors · 2024-12-04
> **Response to Reviewer LdFz's Comments**
>
> We sincerely thank the reviewer for the very positive feedback on our paper. We try to answer the questions as follows.
>
> > Q1: The paper discusses theorems in the main section but only cites others’ contributions. This should be moved to the related works section.
>
> Thank you for the suggestion. We have reorganized our manuscript and included an independent preliminary section to present the previous theoretical building blocks and highlight their relevance to our proposed method. We sincerely appreciate the reviewer’s constructive comment, which has helped us improve the overall organization of our manuscript.
>
> ------
>
>
>
> > Q2: If the algorithm is implemented robustly and scaled to industry infrastructure, it could aid large-scale model training.
>
> We greatly appreciate the reviewer’s positive comment. Our work explores improved performance-efficiency trade-offs, which are particularly meaningful for large-scale model fine-tuning, especially in resource-limited industrial environments. We have added a discussion in the conclusion section to highlight the potential of our work. We sincerely thank the reviewer again for the acknowledgment.
>
> > Q3: The experiments offer a comprehensive comparison of various LoRA baselines.
>
> Thank you for acknowledging our experimental design and results. In response to feedback from other reviewers, we have conducted additional experiments on more complex datasets to further evaluate our method and have reported the consistently superior results in our revised manuscript.

---

### Review · Reviewer_2F9x · 2024-11-22

**Summary Of Contributions:**

This paper aims to improve the performance of the LoRA method (an efficient method for finetuning). First, the authors explain the importance of the intrinsic dimension of $\Delta W = W_{finetuned} - W_{pretrained}$ in the final performance. Then, they claim prior works have two problems: limited LoRA dimension and LoRA finetuning can lead to over/underfitting. To mitigate these problems, they propose two changes to the LoRA blocks: 1) regularization in the weights and 2) random gradient masking.

**Audience:**

Yes

**Claims And Evidence:**

Yes

**Requested Changes:**

* The citation and text have been combined in several locations, such as Page 2 - line 2, Page 3 - line 1, Page 4- line 3 and 14.

* The improvement in the method is insignificant (for example, less than 1 % in Table 1).
   1) The information about random seeds is missing.
   2) LoRA training can be sensitive to learning rate, and using this parameter can change the final observation.

* Some tasks, such as CIFAR100, are very simple and cannot show the importance of finetuning methods in ViT. Authors can include more sophisticated datasets to show the limitations of LoRA and their method's superiority.

* The authors mention various PEFT methods in the related works; however, they have only selected 3 as their baseline. Could you please justify why some of the newer LoRA versionmodifications and algorithms are irrelevant here?

* The authors claim gradient masking can make the training more efficient, but dynamic masking during training is not very beneficial in terms of computation reduction since the gradients are randomly set to zero after calculation. Could you please share any information or evidence that shows the computational superiority of this method?

**Strengths And Weaknesses:**

Strengths:
* The problem is well-motivated.
* The intuition behind the design choices is explained properly.
* Experiments in the NLP domains are more comprehensive.
* The method shows some improvement compared to the prior work.

Weakness:
* Baselines can be improved. The authors have named several works in the prior work sections but only selected two methods for their comparison baseline.
* In some scenarios, the improvement is negligible.
* The method is not very novel. Adding regularization to training parameters is a common practice, and prior works have already explored the benefits of different masking methods on top of the LoRA blocks.

---

> ### Author Response · Authors · 2024-12-04
> **Response to Reviewer 2F9x's Comments**
>
> We sincerely thank the reviewer for the very constructive comments on our paper. We try to address the concerns listed below.
>
> > Q1. The citation and text have been combined in several locations (Page 2 - line 2, Page 3 - line 1, Page 4 - line 3 and 14).
>
> Thank you for pointing this out. We have revised the manuscript to ensure that all citation formats are correct, including both textual and parenthetical citations.
>
> ---
>
>
>
> > Q2. The improvement in the method is insignificant (e.g., less than 1% in Table 1).
>
> Thank you for raising this concern. We acknowledge that the improvement reported in Table 1 may appear modest in absolute terms. However, given the already strong performance of LoRA-based fine-tuning methods, even small improvements are significant, particularly in resource-constrained settings.
>
> For context, similar trends have been observed in other works proposing advancements over LoRA. For instance, in AdaLoRA [1], improvements on certain tasks are also less than 1% (e.g., 0.5% on CoLA, 0.3% on MNLI, and 0.2% on SST-2). Despite these incremental improvements, such methods are considered impactful as they enhance performance while reducing the trainable parameter budget. Our proposed method follows a similar pattern, achieving consistent gains while maintaining computational efficiency.
>
> ------
>
>
>
> > Q3: The information about random seeds is missing.
>
> Thank you for the constructive comment. We have added information about random seeds in Section 5.1 of the updated manuscript. We greatly appreciate the reviewer’s suggestion, which helps improve the reproducibility of our work.
>
> ------
>
>
>
> > Q4: LoRA training can be sensitive to learning rate, which can affect final observations.
>
> Thank you for raising this concern. We conducted experiments with various learning rates and observed that excessively large learning rates degrade test performance, while very small learning rates result in slower convergence and suboptimal results. The learning rate reported in Table 4 of our manuscript was carefully selected to ensure competitive performance across all baselines, aligning with the performance of the DeBERTa-v3-base model as reported in the original DeBERTa paper [2].
>
> ------
>
>
>
> > Q5: Tasks such as CIFAR100 are too simple and do not demonstrate the importance of finetuning methods in ViT. More sophisticated datasets are needed.
>
> Thank you for the constructive suggestion. We have added vision experiments using ViT on the Food-101 dataset and reported the results in Figure 2 of our revised manuscript. The overall trend is consistent with what we observed on the CIFAR100 dataset, further demonstrating the generalizability of our proposed advancement. We sincerely thank the reviewer for this valuable comment, which has helped improve the quality of our work.
>
> ------
>
>
>
> > Q6: The authors mention various PEFT methods but only selected three as their baseline. Please justify this selection.
>
> Thank you for raising this question. Our proposed method is specifically designed as an improvement to LoRA, aiming to explore the upper limits of LoRA-based fine-tuning techniques. Therefore, we focused our comparisons on PEFT methods that are also directly based on LoRA. The selected baselines—LoRA, AdaLoRA, and SoRA—represent state-of-the-art and widely adopted advancements in LoRA-based PEFT. These methods were chosen to ensure a comprehensive comparison with the most relevant and competitive approaches in the field. Following suggestions from other reviewers, we have added further explanations in Section 5.1 of the revised manuscript to clarify the connections among the chosen baselines and provide a stronger justification for our baseline selection.
>
> ------
>
>
>
> > Q7: Gradient masking is claimed to improve efficiency, but dynamic masking during training is computationally expensive. Can you share evidence of its computational superiority?
>
> Thank you for your comment. In our implementation of the proposed gradient masking step, we simplify deployment by masking a portion of the gradients immediately after their calculation, ensuring partial updates to the LoRA parameter matrix, as illustrated in Algorithm 1 of our manuscript. However, in practice, gradient masking does not require the computation of gradients for the masked positions. By skipping these calculations, the backpropagation phase becomes significantly more efficient. Modern training frameworks, such as PyTorch, natively support mechanisms for partial gradient computation, enabling the gradient masking step to be implemented in a highly efficient manner. We have expanded our discussion on the gradient masking step and included additional explanations about its potential computational savings in Section 4.3 of our revised manuscript.
>
> ------

---

> > ### Author Response · Authors · 2024-12-04
> > **References**
> >
> > References:
> >
> > *[1] Qingru Zhang, Minshuo Chen, Alexander Bukharin, Pengcheng He, Yu Cheng, Weizhu Chen, and Tuo Zhao. Adaptive Budget Allocation for Parameter-Efficient Fine-Tuning. In International Conference on Learning Representations, 2023.*
> >
> > *[2] Pengcheng He, Jianfeng Gao, and Weizhu Chen. DeBERTaV3: Improving DeBERTa using ELECTRA-Style Pre-Training with Gradient-Disentangled Embedding Sharing. In International Conference on Learning Representations, 2023.*

---

### Author Response · Authors · 2024-12-04
**Author Responses**

We have made every effort to respond to each reviewer’s comments individually.

We sincerely thank the reviewers for their constructive comments and efforts in helping us improve our manuscript.

---

### Decision · Action_Editor_Dgw9 · 2024-12-25

**Recommendation:** Accept with minor revision

**Comment:**

I recommend accepting the paper with minor revisions. The authors have largely addressed the reviewers' concerns in their responses and revised manuscript. The paper offers valuable insights and a potentially impactful method for improving LoRA.

**Audience:**

Yes, definitely. The paper focuses on improving the efficiency and generalization performance of LORA which is widely used for fine-tuning large pre-trained models. Researchers working on model compression, transfer learning, and resource-constrained machine learning would find these findings valuable. Also, as Reviewer LdFz stated, "If the algorithm is implemented robustly and scale to industry infra, it could help to large scale model training."

**Claims And Evidence:**

The authors generally support their claims with evidence.

Claim 1: Authors state, "This paper investigates the intrinsic dimension of the matrix updates approximated by the LoRA method and reveals the performance benefits of increasing this intrinsic dimension." The authors support this claim through theoretical analysis (Theorem 3.1 citing Zeng & Lee, 2023) and empirical results like in Table 5

Claim 2: Authors state, "The proposed method, termed Regularized and Masked LoRA (RM-LoRA), achieves superior generalization performance with the same or lower trainable parameter budget compared to the original LoRA and its latest variants across various open-source vision and language datasets." Experimental results in Figures 1 and 2 and Tables 1, 2, and 3, comparing their proposed RM-LoRA, R-LoRA, and GM-LoRA with baselines like LoRA, AdaLoRA, and SoRA highlight improvements in metrics like test accuracy and generalization error.